# Intelligent Computation Offloading Mechanism with Content Cache in Mobile Edge Computing

**Feixiang Li [1], Chao Fang [2,\*], Mingzhe Liu [1], Ning Li [1] and Tian Sun [3]**

[1] The 15th Research Institute of China Electronics Technology Group Corporation, Beijing 100083, China
[2] Department of Information and Telecommmunication Engineer, Beijing University of Technology, Beijing 100124, China
[3] Beijing Tsinghua Tongheng Urban Planning and Design Institute, Beijing 100085, China
\* Correspondence: fangchao@bjut.edu.cn

**Abstract:** Edge computing is a promising technology to enable user equipment to share computing resources for task offloading. Due to the characteristics of the computing resource, how to design an efficient computation incentive mechanism with the appropriate task offloading and resource allocation strategies is an essential issue. In this manuscript, we proposed an intelligent computation offloading mechanism with content cache in mobile edge computing. First, we provide the network framework for computation offloading with content cache in mobile edge computing. Then, by deriving necessary and sufficient conditions, an optimal contract is designed to obtain the joint task offloading, resource allocation, and a computation strategy with an intelligent mechanism. Simulation results demonstrate the efficiency of our proposed approach.

**Keywords:** edge computing; resource allocation; computing offloading

## 1. Introduction

Recently, with the explosive increase in traditional mobile cellular networks, the transmission capacities of traditional wireless networks are facing unprecedented challenges, including that the mobile data traffic is increasing with a high growth rate. Furthermore, novel business scenarios in next-generation networks [1–3] continue to increase, placing higher requirements on latency, efficiency, etc. Facing the severe challenges mentioned above, the ultra-dense network (UDN) [4] endows huge access capabilities by providing closer service nodes to users. This network architecture includes macro base stations (MBS) and small cell base stations (SBS). To be specific, relying on the intensive deployment of SBSs, tremendous access capability can be provided for end devices.

Mobile edge computing (MEC) [5,6] is recognized as an effective requirement provider which transfers the computation ability to edge in ultra-dense networks. In multi-user scenario of ultra-dense networks [7,8], it is a major challenge of how to efficiently offload multiple tasks with the constraint of computation, communication, and cache resources, which satisfies the quality of service for users. This offloading problem, established as a mix integer programming (MIP) problem [9], is ubiquitous and complicated in edge cloud system for the following reasons: On the one hand, the tasks offloaded to edge computing system have different characteristics, delay requirement, computation requirement, and energy consumption, etc.; therefore, it is impossible to model this problem accurately. On the other hand, the online tasks are changing dynamically, which requires edge cloud system to make appropriate online decision. Many approaches are proposed to tackle this problem, e.g., convex optimization method, game theory, and heuristic algorithm.

In this paper, we address the computation offloading problem with content cache in ultra-dense networking scenario, i.e., ultra-dense networks. To be specific, a deep reinforcement learning approach [10,11] is employed to help manage task offloading considering the framework of communication, computation, and cache (3C). Numerical results prove that

this novel framework can enhance the profit of MEC efficiently. The main contributions of our paper can be summarized as follows.

- We focus on the computation offloading problem with content cache in mobile edge computing. Specifically, edge clouds are denoted as service providers and user equipment (UEs) are requestors. With the assistance of multiple edge clouds, this network architecture tends to improve the effectiveness of computation offloading service.
- Considering the framework of communication, computation, and cache, we establish the computation offloading problem model aided by deep Q-learning algorithm. Offloading decisions are modeled as actions in this approach, which in efficient decision action space. Furthermore, we applied deep Q-learning to choose the reasonable approach in solving resource management problem.
- Extensive simulation validates the effectiveness of the proposed approach in ultra-dense networks. Numerical results demonstrate its well performance with the increasing iterations. Additionally, the impact of different parameters in this approach is analyzed.

This paper has the following organization: Section 2 describes related work. The system model is presented in Section 3. Intelligent computation offloading mechanism with content cache is described in Section 4. Section 5 contains the analysis of the experimental results. Section 6 provides the conclusion.

## 2. Related Work

There are many related works that jointly model the computation offloading problem and resource allocation problem in MEC networks as the MIP problems. Ref. [12] considered the task computation offloading problem in 5G cellular networks as an energy consumption model, which satisfied the constraints of computing capacity and service delay. Ref. [13] utilized dynamic voltage adjustment to optimize the computation speed and transmission power of UEs. Aiming at two main objectives, i.e., energy consumption and time delay, the authors transformed this non-convex problem into a convex problem by variable substitution method. Ref. [14] focused on the multiuser computation offloading problem in the presence of multiple channel interference. Coefficients were proposed to weigh the energy consumption and execution delay of the offloading problem. The authors proposed a distributed computation offloading algorithm to achieve Nash equilibrium. Ref. [15] analyzed energy consumption and time delay in the multiple terminal devices scenario. In this scenario, the authors considered optimizing communication and computing resources between edge clouds and terminal devices. Ref. [16] focused on computation offloading decision and computational resource allocation. The edge cloud made decisions relying on not only the computation requirement of the whole terminal devices but also the computation resources of servers. Ref. [17] proposed a novel online decision making approach to determining the pre-processing level for either higher result accuracy or better energy efficiency in a mobile environment. Ref. [18] proposed a Lyapunov based on-line approach designation mechanism that dynamically chooses an appropriate data communication approach based on data transmission queue, estimated network conditions, and the device moving speed. The aforementioned works employ traditional methods to solve the computation offloading problem, which is difficult to adapt to the dynamic change in the novel wireless network scenario.

Motivated by the success of artificial intelligence (AI) in a wide spectrum of fields, it is envisaged that AI powered edge computing could overcome the emerging challenges by fully unleashing the potential of the edge big data. Ref. [19] developed a deep reinforcement learning approach to tackle the edge caching and computing problem in vehicle networks. In addition, the authors proposed the mobility-aware reward to improve the efficiency of vehicular networks. Ref. [20] leveraged a deep reinforcement learning algorithm to learn the optimal computation offloading and packet scheduling policies to solve the multi-tenant cross-slice radio resource orchestration problem. In mobile social networks [21], the authors applied a novel deep reinforcement learning approach to automatically make a

decision for optimally allocating the network resources under the framework of computing, caching, and communication (3C). Ref. [22] proposed a Deep Reinforcement learning-based Online Offloading (DROO) framework from the past offloading experiences under various wireless fading conditions and automatically improved its action generating policy. This framework was verified in order to greatly reduce the computational complexity, especially in large-size networks. The above references proposed DRL in computation offloading problem; however, they do not focus on the ultra-dense network scenario, which consists of multiple edge clouds and multiple UEs. Therefore, in this paper, we focus on the computation offloading problem in a novel scenario, i.e., an ultra-dense network. Furthermore, we propose a DRL-aided computation offloading scheme to achieve resource allocation automatically.

## 3. System Model

### 3.1. Architecture of Mobile Edge Computing

From Figure 1, the architecture of mobile edge computing is formed by a macro base station (MBS), a small base station (SBS), and user equipment (UE). Specifically, MBS and SBS are able to provide computing service using their edge clouds. The MBS edge cloud owns a powerful calculation capability while the SBS edge cloud has a low computation delay. UEs are capable of choosing different offloading computation schemes according to its requirement.

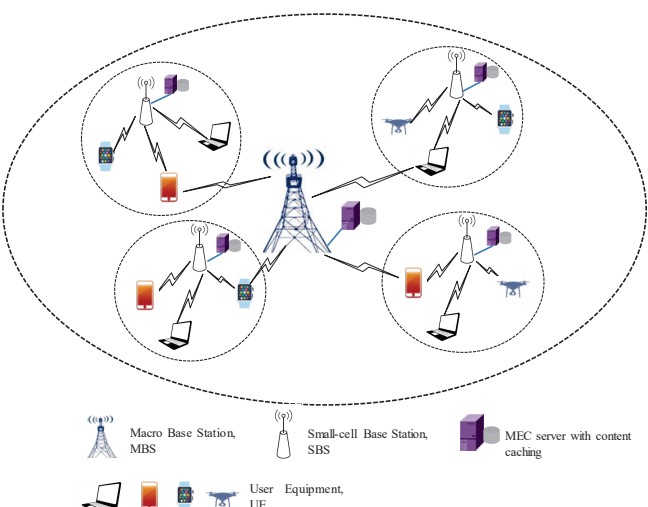

**Figure 1.** Architecture of mobile edge computing.

From Figure 2, the computation offloading service between UEs and edge cloud includes three steps. First, the task in the UE is chosen to be offloaded to either SBS edge cloud or MBS edge cloud. Second, the edge cloud chooses a decision on the execution of the offloaded task. If the task has not been cached in the edge cloud, it will be computed. Otherwise, the cached result will be returned directly. Third, the edge cloud returns the executed result back to UEs.

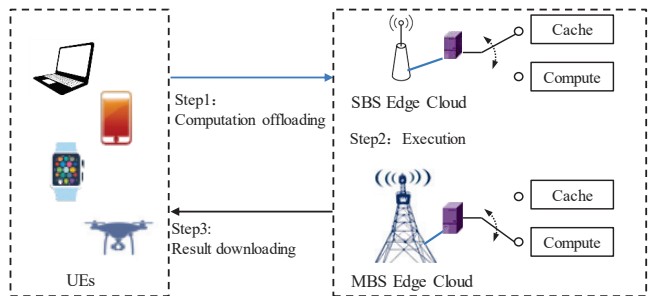

**Figure 2.** Computation offloading with caching in ultra-dense networks.

In this paper, we focus on the offloading decision and resource management problem in computation offloading. Offloading decision is trying to find an optimal decision strategy when considering the framework of communication model, computation model, and caching model in MEC. In addition, resource management addresses the minimal queue delay in the servers of edge cloud. To conclude, deep reinforcement learning algorithm is utilized to solve this problem.

*3.2. Communication Model*

In this scenario, $\mathbf{N} = \{1, 2, \cdots, N\}$ is set as SBSs. UEs are denoted by $\mathbf{M} = \{1, 2, \cdots, M\}$. $J_m = \{A_m, A'_m, B_m, T_m^{max}\}$ represented the task from UE $m$. To be specific, $A_m$ and $A'_m$ denote the volume of the task $J_m$ before and after calculation. $B_m$ means CPU number required by task $J_m$. For task $J_m$, $T_m^{max}$ stands for the maximum delay. Task $J_m$ chooses the following two schemes base on its requirement.

(1) Task offloading scheme by MBS. In this scheme, $w_{m0}$ is represented as the channel bandwidth between UE $m$ ($m \in \mathbf{M}$) and MBS . Moreover, $\sigma^2$ is the constant addictive noise power and $p_{m0}$ means the transmission power. Additionally, $h_{m,0}$ represents the channel gain and $k_{m0} \in \{0, 1\}$ is the interference factor. Therefore, the data transmission rate $r_{m0}$ can be formulated as

$$r_{m0} = w_{m0} \log_2 \left( 1 + \frac{p_{m0}|h_{m0}|^2}{\sigma^2 + \Sigma_{m=1}^{M} k_{m0} p_0 |h_{m0}|^2} \right). \tag{1}$$

In this model, we assume that $\alpha_{m0}$ represents the access fee from UE $m$, charged by the MBS edge cloud. $\beta_{m0}$ is defined as the usage cost of spectrum, paid by the MBS edge cloud. Therefore, the communication revenue can be formulated as $R_{m0}^{comm} = \alpha_{m0} r_{m0} - \beta_{m0} w_{m0}$. To be specific, $\alpha_{m0} r_{m0}$ is the income of the MBS edge cloud from user $m$, and $\beta_{m0} w_{m0}$ is cost of the MBS edge cloud to pay for the usage of bandwidth.

(2) Task offloading scheme by SBS. Similar to the task offloading scheme by MBS, the channel bandwidth is shown by $w_{mn}$ and $p_{mn}$ is the transmission power. Furthermore, $\sigma^2$ means the constant addictive noise power and $h_{m,n}$ represents the channel gain. Therefore, the data transmission rate $r_{mn}$ is formulated as

$$r_{mn} = w_{mn} \log_2 \left( 1 + \frac{p_{mn}|h_{mn}|^2}{\sigma^2 + \Sigma_{m=1}^{M} k_{mn} p_n |h_{mn}|^2} \right), \tag{2}$$

where $k_{mn} \in [0, 1]$ represents the interference between UE $m$ and the SBS edge cloud $n$.

Suppose $\alpha_{mn}$ represents the access fee paid by UE $m$. The usage cost of spectrum is denoted as $\beta_{mn}$, which is paid by the SBS edge cloud $n$. Similarly, the communication revenue can be formulated as $R_{mn}^{comm} = \alpha_{mn} r_{mn} - \beta_{mn} w_{mn}$. Specifically, $\alpha_{mn} r_{mn}$ indicates the income of the SBS edge cloud $n$ from user $m$, and $\beta_{mn} w_{mn}$ is cost of the SBS edge cloud $n$ to pay for the usage of bandwidth.

### 3.3. Computation Model

(1) Computation model for MBS. The *computation delay* for MBS is calculated as $T^c_{m0} = \frac{B_m}{C_{m0}}$. Specifically, $C_{m0}$ stands for the calculation capacity of MBS.

The calculation rate of task $m$ by MBS is deemed as

$$q_{m0} = \frac{A_m}{B_m / C_{m0}} = \frac{A_m C_{m0}}{B_m},\tag{3}$$

The *computation energy consumption* is represented as $e_{m0} = \varpi_{m0} T^c_{m0}$, where $\varpi_{m0}$ is denoted as the energy consumption per second for the MBS edge cloud.

Assume $\phi_{m0}$ represents the computation fee from user $m$, charged by MBS. Moreover, MBS pays for the computation cost to calculate the task in the MBS edge cloud is denoted as $\varphi_{m0}$. The computation revenue model can be established as $R^{comp}_{m0} = \phi_{m0} q_{m0} - \varphi_{m0} e_{m0}$. Specifically, $\phi_{m0} q_{m0}$ denotes the income of the MBS edge cloud from user $m$, and $\varphi_{m0} e_{m0}$ is the cost of the MBS edge cloud to pay for the usage of servers.

(2) Computation model for SBS. Similarly, the *computation delay* executed by SBS $n$ for task $J_m$ is represented as $T^c_{mn} = \frac{B_m}{C_{mn}}$, where $C_{mn}$ is the calculation capacity of SBS $n$.

The calculation rate of task $m$ by SBS $n$ is expressed as

$$q_{mn} = \frac{A_m}{B_m / C_{mn}} = \frac{A_m C_{mn}}{B_m},\tag{4}$$

To be specific, the *computation energy consumption* by SBS $n$ is represented as: $e_{mn} = \varpi_{mn} T^c_{mn}$, where $\varpi_{mn}$ is the energy consumption per second for SBS $n$.

Suppose the computation fee charged by SBS $n$ is denoted as $\phi_{mn}$. The computation cost $\varphi_{mn}$ is defined to calculate the task in the SBS $n$, which is paid by SBS $n$. Eventually, the computation revenue model can be established as $R^{comp}_{mn} = \phi_{mn} q_{mn} - \varphi_{mn} e_{mn}$. Meanwhile, $\phi_{mn} q_{mn}$ indicates the income of SBS $n$ from user $m$, as well as $\varphi_{mn} e_{mn}$ is cost of SBS $n$ to pay for the usage of servers.

### 3.4. Caching Model

In this scenario, we consider $D$ contents are requested in both edge clouds. The caching strategy is determined by the binary parameter $x'$, $x' = 1$ means the content is cached in the edge cloud, while $x' = 0$ represents not.

Furthermore, the content popularity distribution is denoted with $\mathbf{G} = \{g_1, g_2, \cdots, g_D\}$, where $D$ is the maximal type number of content. Additionally, each UE requests the content $d$ with the probability $g_d$. In general, $\mathbf{G}$ submits to Zipf distribution [23], and can be formulated as

$$g_d = \frac{1/d^\epsilon}{\Sigma^D_{d=1} 1/d^\epsilon},\tag{5}$$

where the content popularity is characterized by $\epsilon$, and its range is [0.5,1.5] [24]. Then, the gain is formulated as

$$l_{A'_m} = \frac{g_{A'_m} A'_m}{T_{A'_m}},\tag{6}$$

where $T_{A'_m}$ is the time for downloading cached contents. In this paper, the price for caching the contents was already known in advance. The backhaul cost is defined as $\gamma_{m0}$, which is paid by MBS. Furthermore, $\psi_{m0}$ means the storage fee to cache the content $A'_m$ by MBS, which is charged by MBS. To conclude, the caching revenue of the MBS edge cloud can be established as $R^{cache}_{m0} = \psi_{m0} l_{A'_m} - \gamma_{m0} A'_m$. Specifically, $\psi_{m0} l_{A'_m}$ means the income of MBS from UE $m$, and $\gamma_{m0} A'_m$ deems the cost of MBS to pay for the usage of backhaul bandwidth.

Similarly, let $\gamma_{mn}$ denote the backhaul cost of SBS $n$. Moreover, the storage fee at SBS $n$ is represented as $\psi_{mn}$. Therefore, the caching revenue of the SBS edge cloud $n$ can be calculated as $R_{mn}^{cache} = \psi_{mn}l_{A'_m} - \gamma_{mn}A'_m$. To be specific, $\psi_{mn}l_{A'_m}$ is the income of SBS $n$ from UE $m$, and $\gamma_{mn}A'_m$ represents the cost for SBS $n$ to pay to the usage of backhaul bandwidth.

## 4. Deep Reinforcement Learning

### 4.1. Reinforcement Learning Algorithm

The reinforcement learning algorithm simulates the interaction between an agent and the environment. An agent can obtain observation from the environment and adopt action. Afterwards, the environment will return a reward to the agent. To be specific, reinforcement learning focuses on a multi-step decision-making problem, which tries to achieve a goal through the multi-step appropriate decision in a changing situation. Different from other machine learning algorithms, reinforcement learning does not need to learn from the experienced samples. Instead, it is capable of obtaining feedback from its attempt action. Reinforcement learning includes four parts: experiment states, actions, rewards, and the probability of state transitions. In conclusion, the reinforcement learning algorithm is a powerful method to solve real-world problems without prior knowledge.

According to interactions with the environment, the Q-learning algorithm attempts to find the optimal behavior through continuous attempts. The optimal behavior concerns not only immediately reward, but also the reward of any following steps in the future. The decision process of Q-learning algorithm is based on the Markov decision process, and it can be expressed by a quintuple: $\{s_i, a_i, P(s_i, a_i, s_{i+1}), R(s_i, a_i), Q(s_i, a_i)\}$. Meanwhile, $s_i$ denotes the state space, and $a_i$ is termed as the action space. In addition, the probability $P(s_i, a_i, s_{i+1})$ helps to choose action $a_i$ to transmit state $s_i$ to the next state $s_{i+1}$. $R(s_i, a_i)$ represents the immediate reward when the system in state $s_i$ chooses the action $a_i$. $Q(s_i, a_i)$ is the cumulative reward value at the condition of when action $a_i$ is chosen for state $s_i$.

Supposed that the state in step $i$ is denoted as $s_i$, the reward of each state is presented as $V^\pi(s_i)$, and its function is formulated as

$$V^\pi(s_i) = R_i + \lambda R_{i+1} + \lambda^2 R_{i+2} + \cdots , \tag{7}$$

where $R_i$ is the reward in step $i$, $\lambda$ ($0 < \lambda < 1$) represents the influence for the training agent to the future reward. In this algorithm, Q value is the estimate of state and action, and its formulation is

$$Q(s_i, a_i) = R_i + \lambda V^\pi(s_{i+1}). \tag{8}$$

Relying on the formulations above, $Q_{i+1}(s_i, a_i)$ is updated by (9). Specifically, the learning rate $\eta(\eta \in (0, 1))$ controls the learning speed.

$$Q_{i+1}(s_i, a_i) = (1 - \eta)Q_i(s_i, a_i) + \eta[R_i + \lambda max Q_i(s_{i+1}, a_{i+1})]. \tag{9}$$

### 4.2. Deep Q-Network

From Algorithm 1, the idea of deep Q-network [25] is utilizing a feedforward artificial neural network to approximate the Q-value function $Q(s, a; \theta)$. The input layer of this Q-network is the state $s$. The output layer is Q-values corresponds to action $a$ taken at state $s$. The parameter $\theta$ via small steps that minimize a loss function:

$$L(\theta) = E[(y(s, a, s'; \hat{\theta}) - Q(s, a; \theta))^2] \tag{10}$$

where the target function $y(s, a, s'; \hat{\theta}) = R + \lambda max Q(s', a'; \hat{\theta})$ changes when the parameter $\hat{\theta}$ are updated.

---

**Algorithm 1** Deep Q-learning Algorithm

---

**Input**: state $s$, action $a$;
**Output**: $Q(s,a)$;
**Initialization**:
Initialize the main deep Q-network with weight $\theta$;
**for** $i < T$ **do**
  **if** random probability $P < \delta$ **then**
    select an action $a_i$;
    **otherwise**
    $a_i = argmax\ Q(s,a;\theta)$;
  **end if**
  Execute action $a_i$ in the system, then obtain the reward $R_i$ and the next observation $s_{i+1}$;
  Calculate the target Q-value
  $y(s,a,s';\hat{\theta}) = R + \lambda max\ Q(s',a';\hat{\theta})$;
  Update the main deep Q-network by minimizing the loss $L(\theta)$ with (10);
**end for**

---

## 5. Deep Q-Learning Aided Computation Offloading with Content Cache

In the ultra-dense network scenario, the uplink channel conditions and computation capabilities are changing dynamically. It is difficult to employ traditional methods to find an optimal solution. In contrast, DRL does not require a well-captured model or prior information. It is capable of adaptively refining the strategy from the environment. Therefore, we introduce deep Q-learning to find the optimal action effectively, as shown in Figure 3. To be specific, we propose different strategies according to whether or not the offloaded task is cached in the edge cloud. On the one hand, if the offloaded task is cached, the edge cloud should consider the caching model to save the computation delay. On the other hand, if the offloading task is not cached, the edge cloud will calculate this task directly.

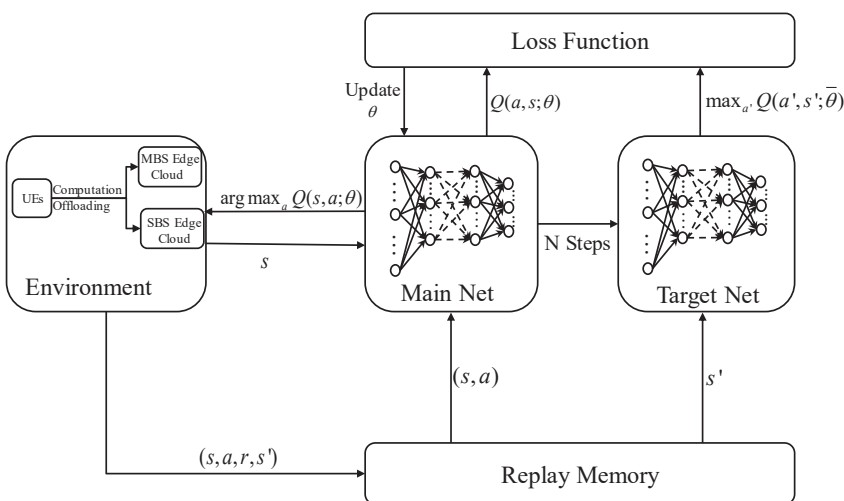

**Figure 3.** Deep reinforcement learning for computation offloading with content cache.

State Space. The state of the edge cloud $n \in \mathbf{N}$ and the available cache $d \in \mathbf{D}$ for UE $m \in \mathbf{M}$ in timeslot $t$ is determined by the random variables $a_m^{comm}$ and the random variables $a_m^{cache}$. In addition, the input data also embraces the contact number and contact time per timeslot for each of the UEs and edge clouds, respectively.

Action Space. In this edge cloud system, the agent should determine where this task is offloaded according to the limited communication resource, whether or not the offloaded task has been cached in the server.

Accordingly, the current action $a_m(t)$ at timeslot $t$ is defined as

$$a_m(t) = \{a_m^{comm}(t), a_m^{cache}(t)\}, \tag{11}$$

where $a_m^{comm}(t)$ and $a_m^{cache}(t)$ are represented as follows.

First, we define row vector $a_m^{comm}(t) = [a_{m,1}^{comm}(t), a_{m,2}^{comm}(t), \cdots, a_{m,N}^{comm}(t)]$, where $a_{m,i}^{comm}(t), i \in \{1, 2, \cdots, N\}$ denotes whether UE $m$ are connected to edge cloud. To be specific, the value of $a_{m,i}^{comm}(t)$ is $\{0, 1\}$, where $a_{m,i}^{comm}(t) = 0$ means that at timeslot $t$ the task in UE $m$ choose to be offloaded to MBS, otherwise $a_{m,i}^{comm}(t) = 1$ means that at timeslot $t$ the task in UE $m$ choose to be offloaded to SBS.

Then, we define row vector $a_m^{cache}(t) = [a_{m,1}^{cache}(t), a_{m,2}^{cache}(t), \cdots, a_{m,N}^{cache}(t)]$, where $a_{m,j}^{cache}(t), j \in \{1, 2, \cdots, N\}$ denotes whether the content of UE $m$ has been cached. To be specific, the value of $a_{m,j}^{cache}(t)$ is $\{0, 1\}$, where $a_{m,j}^{cache}(t) = 0$ means that at timeslot $t$ the content is not cached, otherwise $a_{m,j}^{cache}(t) = 1$ means that at timeslot $t$ the content is cached.

*Reward function.* The reward of this edge cloud system is to maximize the revenue of communication model, computation model, and caching model. The reward function for UE $m$ is defined

$$R_m(t) = R_m^{comm}(t) + R_m^{cache}(t), \tag{12}$$

$R_m^{comm}(t)$ represents the reward from the communication model for UE $m$, which includes the communication revenues of both edge clouds. Moreover, it can be formulated as

$$
\begin{aligned}
R_m^{comm}(t) &= (1 - a_m^{comm}(t))R_{m0}^{comm} + a_m^{comm}(t)R_{mn}^{comm} \\
&= (1 - a_m^{comm}(t))(\alpha_{m0}r_{m0} - \beta_{m0}w_{m0}) + \\
&\quad a_m^{comm}(t)(\alpha_{mn}r_{mn} - \beta_{mn}w_{mn}).
\end{aligned} \tag{13}
$$

$R_m^{cache}(t)$ represents the reward from whether or not the content is cached, which includes the computation revenues and cache revenues of both edge clouds. Moreover, it is formulated as

$$
\begin{aligned}
R_m^{cache}(t) &= (1 - a_m^{comm}(t))a_m^{cache}(t)R_{m0}^{cache} + (1 - a_m^{comm}(t)) \\
&\quad (1 - a_m^{cache}(t))R_{m0}^{comp} + a_m^{comm}(t)a_m^{cache}(t)R_{mn}^{cache} \\
&\quad + a_m^{comm}(t)(1 - a_m^{cache}(t))R_{mn}^{comp}. \\
&= (1 - a_m^{comm}(t))a_m^{cache}(t)(\psi_{m0}l_{A'_m} - \gamma_{m0}A'_m) + \\
&\quad (1 - a_m^{comm}(t))(1 - a_m^{cache}(t))(\phi_{m0}q_{m0} - \varphi_{m0}e_{m0}) \\
&\quad + a_m^{comm}(t)a_m^{cache}(t)(\psi_{mn}l_{A'_m} - \gamma_{mn}A'_m) + \\
&\quad a_m^{comm}(t)(1 - a_m^{cache}(t))(\phi_{mn}q_{mn} - \varphi_{mn}e_{mn}).
\end{aligned} \tag{14}
$$

Meanwhile, if the content is cached, the strategy chooses the revenue of the caching model, otherwise, the content has to be calculated in the edge cloud.

## 6. Deep Q-Learning Aided Resource Management in Edge Cloud

In this section, we will focus on the intelligent resource management for computation service in edge clouds. In the multi-user scenario, edge clouds which are deployed on the mobile access network will provide computation service for all users within its range simultaneously. In the user intensive area, when there are many user devices using the computing service, edge cloud may not be able to process the migrated tasks and return the results in time due to resource constraints and other reasons, resulting in large task processing delay. In such an environment, if user equipment (UE) runs virtual reality, ultra-high definition video, or other applications that are very sensitive to time delay, it will greatly affect QoS, which requires edge cloud to comprehensively consider the sensitivity of computing tasks to time delay and the demand of tasks for system resources to allocate computing resources and provide computing services according to priority.

In this section, aiming at the task offloading problem in the above multi-user MEC scenarios, considering the limitation of edge cloud computing resources when UE computing power is low and all tasks need to be migrated to edge cloud for execution, the system model and task model are constructed, and the optimization objectives of minimizing the average delay and average overtime of tasks are formulated. On this basis, this paper proposes a task transfer strategy based on deep reinforcement learning. By constructing MECs's computing resource allocation process into Markov decision-making process, the task transfer problem of multi-user MEC is transformed into a strategy learning problem. After a period of strategy learning using deep reinforcement learning algorithm, this strategy can improve the overall effectiveness of MEC system rate and service quality of users.

### 6.1. System Model and Task Model

Edge cloud in MEC is a multiple computer cluster. In this subsection, the cluster is modeled as a super server with many types of resources (CPU, memory, input and output devices, etc.), in which task scheduler has a complete view of the list of all computer computing resources in the cluster and their usage, and ignores the impact of fragmentation effects such as the underlying machine communication in the cluster. Although this modeling ignores the impact of many underlying machine communication in the actual scheduling of computer clusters, it retains the basic elements in the multi task and multi resource scheduling process of server clusters, greatly reducing the complexity of the model and simplifying the algorithm design.

Figure 4 shows the modeled MECs system structure diagram, which includes a data transmission unit, a task cache queue, a task scheduler with all the resource usage views of MECs ,and a logic execution unit. The data transmission unit is mainly responsible for receiving the time delay sensitivity calculation task and returning the task calculation result in UE; the task cache queue is mainly used to cache the received tasks to be processed and wait to be scheduled by the task scheduler; the task scheduler is the brain of the whole system, which is responsible for the resource management (allocation, recovery, etc.) of the server cluster system and the task scheduling in the task cache queue; and the logical execution unit is the resource pool of the system, which contains all the computing resources related to task processing in the cluster. This section assumes that the resource requirements of a task have been determined when the task arrives. Therefore, the complete process of processing a calculation task in a multi-user MEC system is as follows: UE sends task data to MECs through a wireless channel. When MECs transmission unit receives the calculation task, it first stores the task in the MECs task cache queue (at this time, the resource requirements of the task have been determined). It must wait for the task scheduler to allocate resources for the system according to the occupancy of resources and schedule the task into the logical execution unit for execution. After the task is executed, the calculation result is obtained and the result is returned to the corresponding UE through the transmission unit.

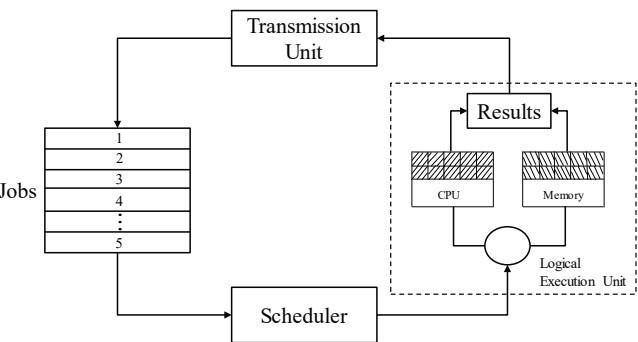

**Figure 4.** Architecture of servers in MEC.

In this section, we denote $J$ as the jobs which queue in MEC servers. Assume that the number of jobs in MEC servers is $N$, then jobs can be expressed as $J = \{J_1, J_2, \cdots, J_i, \cdots, J_N\}$, where $J_i$ indicates the $i$th job in the MEC servers. The resources which should be occupied by $J_i$ have been determined in advance, and the resources will be recycle allocated after execution. To be specific, $J_i$ is denoted as $J_i = \{t_i^{enter}, t_i^{start}, t_i^{end}, R_i, t_i^{proc}, t_i^{total}\}$. Meanwhile, $t_i^{enter}$ represents the time when $J_i$ enters the queue in MEC servers. $t_i^{start}$ indicates the time when $J_i$ is scheduled, and $t_i^{end}$ means the time when $J_i$ is executed. Moreover, $R_i$ represents the required resources by job $J_i$, which can be denoted as $R_i = \{C_i, M_i\}$. In detail, $C_i$ means the required number of CPU units, and $M_i$ indicates the number of memory units required by job $J_i$. In addition, $t_i^{proc}$ means the processing time in MEC servers by job $J_i$. $t_i^{total}$ indicates the total time in MEC servers by job $J_i$, which includes queue time and processing time. The concrete description is shown in Figure 5.

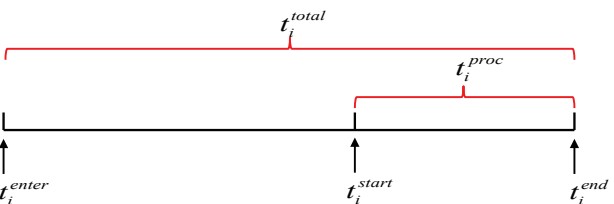

**Figure 5.** Description of different times.

The global view of task scheduler in MECs contains different amounts of CPU resources and memory resources, which can process multiple computing tasks in parallel. In the case of sufficient resources, in order to ensure the timeliness of tasks, the task scheduler will schedule all tasks into the execution unit for execution; at this time, there are no tasks waiting to be executed in the task cache queue. When the number of tasks migrated to MECs increases, and its resources can not meet the needs of all tasks, MECs will cache them first. According to the waiting time, timeout time and resource demand table of tasks, MECs will select the task scheduling order that makes the overall performance of MEC system optimal to allocate computing resources.

In this section, in order to optimize MECs task scheduling and processing as a whole, the optimization objectives will be modeled relying on two aspects: task average time delay and task average over time.

(1) Task average time delay. For job $J_i$, in order to ensure the fairness of scheduling tasks in MECs task scheduler and take into account the performance of the system, we use the ratio of task completion delay to task processing delay as the optimization goal of the task. The expression is established as

$$k_i = \frac{t_i^{end} - t_i^{enter}}{t_i^{proc}}, \tag{15}$$

where $t_i^{end} - t_i^{enter}$ means the minus value between the task completion time and the time when the task enters the queue, indicating the completion delay from the task entering MECs to the execution completion. Thus, the definition of average time delay ratio can be modeled as

$$K = \frac{1}{I} \Sigma_{i=1}^{I} k_i. \tag{16}$$

When using the average time to delay ratio as the optimization goal, for the tasks to be executed in the task cache queue, if the current system resources meet the running requirements of the task, the task scheduler is more inclined to schedule the tasks with short running time, and can also take into account the tasks with long running time to ensure that there is no starvation effect.

(2) Task average over time. For job $J_i$, the average over time is established as

$$o_i = max\{0, t_i^{end} - t_i^{enter} - t_i^{total}\}. \tag{17}$$

From (17), $o_i = 0$ indicates that the current job is not currently timed out, and the over time is set as 0. Otherwise, $o_i > 0$ represents that when the current job is timeout, the overtime is the value of $o_i$. Similarly, to minimize the overtime of jobs generally, we set the average overtime as the final optimization objective, and its formulation is established as

$$O = \frac{1}{I}\Sigma_{i=1}^I o_i. \tag{18}$$

Therefore, the whole optimization objective of this section is to minimize the weighted sum of the average time delay ratio and the average overtime, and the formulation is

$$\min \mu_1 K + \mu_2 O, \tag{19}$$

where $\mu_1$ and $\mu_2$ are weight factors, which submit to the following conditions.

$$\begin{aligned} \mu_1 + \mu_2 &= 1, \\ 0 \le \mu_1 &\le 1, \\ 0 \le \mu_2 &\le 1. \end{aligned} \tag{20}$$

### 6.2. Design of Task Schedule Strategy in Mobile Edge Cloud

In this section, according to Markov decision-making process, task transfer strategy based on deep reinforcement learning is designed from state space, action space, immediate return, and function approximator. The figure of state space is presented in Figure 6.

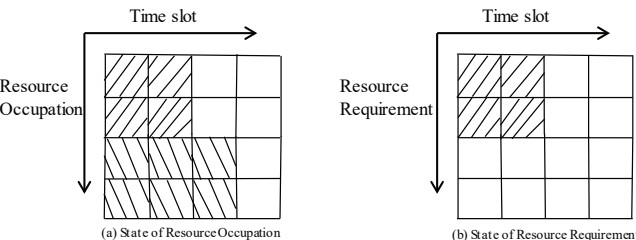

**Figure 6.** Description of State Space.

(1) State Space. The state space is defined as the time distribution of the usage state of each computing resource and the resource demand state of the task in the task cache queue. The resources in MECs system include CPU resources and memory resources. As shown in Figure 6a, the system resource usage status is a state matrix composed of the current time step and the resource usage status of the future time step system. Different lines in the figure represent different tasks. Assuming that the resources represented in the current figure are CPU resources, then the left line represents tasks in the current and future time step. It needs to occupy two units of CPU and execute three time slots. The right line means that it needs to occupy two units of CPU and execute three time steps. Figure 6b shows the estimated resource requirements of tasks in the task cache queue in the current and future time steps, in which left line tasks are expected to occupy two units of resources and maintain two time steps.

In order to solve the problem of fixed input dimension in deep reinforcement learning, only the state of the first D tasks in the task cache queue and MECs resource usage state are taken as the state space during algorithm training. Because the tasks in the task cache queue are arranged in the order of their arrival, a reasonable strategy will give priority to the tasks with long waiting time, which is also consistent with this section. The

optimization objectives are consistent. In addition, the MECs state space is modeled as a matrix so that the deep reinforcement learning algorithm can make full use of its spatial correlation for feature extraction when training and using, which makes the extracted features more representative.

(2) Action Space. According to the system resource state and task resource demand state, the task scheduling unit is defined to allocate the required resources for the task and schedule it as an action, so the action space can be expressed as:

$$A = \{\phi, 1, 2, \cdots, i, \cdots, D\} \tag{21}$$

where $\phi$ represents the action in current time slot is empty, i.e., task scheduling does not schedule the tasks in the task cache queue due to insufficient system resources. Choosing action $i$ means the $i$th task in the task queue is allocated computation resource, and $D$ is depth of action space.

Considering that MECs can provide auxiliary computing services for all tasks in its radiation range at the same time, and it can process tasks in parallel according to the occupancy of resources, so in each time step, reasonable actions should be able to allocate resources and schedule multiple tasks at the same time according to the occupancy of resources, but this will greatly increase the complexity of action space It increases the difficulty of convergence. In order to solve this problem, this section adopts the strategy of action and time decoupling in the process of algorithm implementation: when the task scheduler schedules a task correctly, it freezes the current time step and schedules the next task. Only when the task scheduler selects the empty action or the invalid action (MECs available computing resources can not meet the resource requirements of the task) will the time step to move forward. In this way, it can not only meet the needs of scheduling multiple tasks at the same time, but also reduce the complexity of action space and the difficulty of algorithm training.

(3) Immediate reward. The optimization goal of this section is to reduce the average time delay ratio of MECs task queue and the average time-out length of tasks through reasonable calculation resource allocation, so as to improve the efficiency of MEC system and enhance the user experience. To guide the strategy toward this goal, the immediate reward is modeled as

$$R = \mu_1 \times R_t + \mu_2 \times R_r \tag{22}$$

From the above equation, $\mu_1$ and $\mu_2$ are the discounting factors in the reward, and $R_t$ indicates the immediate reward relying on processing delay, and it can be formulated as

$$R_t = \Sigma_{i=1}^{I} \frac{1}{k_i}. \tag{23}$$

$R_r$ represents the immediate reward from overtime, and it can be modeled as

$$R_r = \begin{cases} 0, o_i = 0; \\ \frac{1}{o_i}, o_i > 0. \end{cases} \tag{24}$$

In the case of $o_i = 0$, at this time the task does not over time, and the immediate return reaches the maximum. In this way, the algorithm can be guided to learn a computing resource allocation strategy to reduce the time-out of the task in the way of reinforcement learning. In the case of $o_i > 0$, the task timeout, the longer the timeout, the larger the immediate reward.

The values of discounting factors place an important role in learning algorithm, if $\mu_1 = 1, \mu_2 = 0$, maximizing the cumulative discounted return is equivalent to minimizing the average time delay ratio. If $\mu_1 = 0, \mu_2 = 1$, maximizing the cumulative discounted return is equivalent to minimizing the average over time. Therefore, the immediate reward designed in this section is to optimize and minimize the tradeoff between the average time delay ratio and the average overtime, and $\mu_1$ and $\mu_2$ are set as 0.5, respectively.

(4) Function approximator. In this section, convolution neural network is used as a function approximator. Compared with full connection artificial neural network, convolution neural network increases the consideration of spatial correlation, and can extract high-level semantic information more efficiently, making the extracted features more representative. Figure 7 shows the structure of convolutional neural network. The input of the network is the state space model; the hidden layer is convolutional neural network and full connection layer. Softmax function is used to output the probability distribution of action space. In order to increase the nonlinearity of the network, relu is used as the activation function.

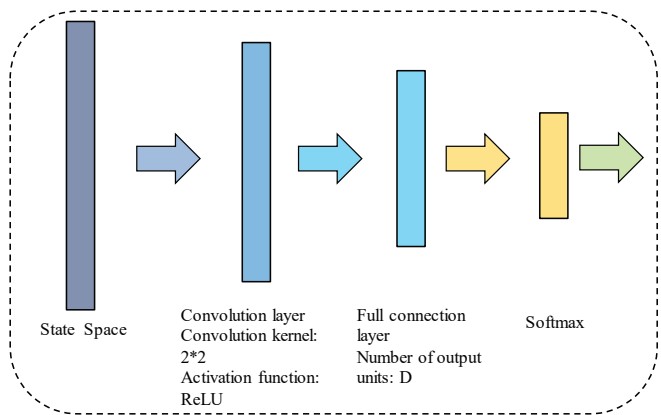

**Figure 7.** Convolution Neural Network.

## 7. Numerical Results

In this section, the computing offloading with content cache mechanism based on deep reinforcement learning will be simulated and evaluated on Spyder. Subsection A describes the simulation experiment parameters and the experimental results are shown in subsection B. Furthermore, the influence of the parameters of the algorithm is discussed.

### 7.1. Parameter Setting

The simulation scenario is shown in Figure 8 and the parameters in this experiment are presented in Table 1.

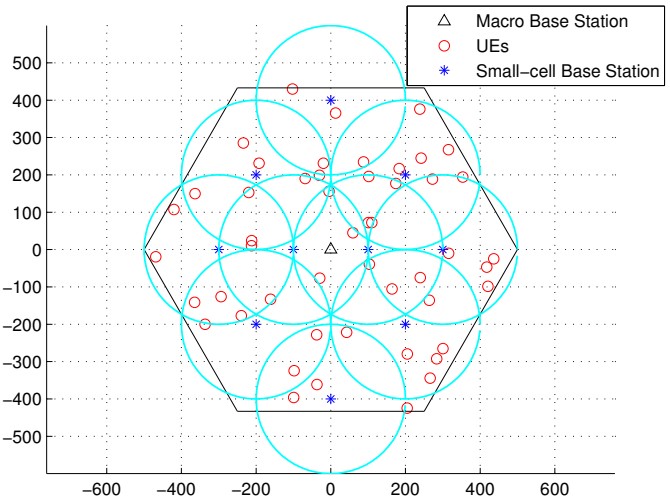

**Figure 8.** Simulation scenario.

**Table 1.** Parameter settings.

| System Parameters | Value Setting |
| --- | --- |
| The access fee charged by MBS edge cloud | [3, 6] units/bps |
| The access fee charged by SBS edge cloud | [1, 3] units/bps |
| The usage cost of spectrum paid by MBS edge cloud | $[3 \times 10^{-4}, 6 \times 10^{-4}]$ units/Hz |
| The usage cost of spectrum paid by SBS edge cloud | $[1 \times 10^{-4}, 3 \times 10^{-4}]$ units/Hz |
| The computation fee charged by MBS edge cloud | 0.8 units/J |
| The computation fee charged by SBS edge cloud | 0.4 units/J |
| The computation cost paid by MBS edge cloud | 0.2 units/J |
| The computation cost paid by SBS edge cloud | 0.1 units/J |
| The storage fee charged by MBS edge cloud | 20 units/byte |
| The storage fee charged by SBS edge cloud | 10 units/byte |
| The backhaul cost paid by MBS edge cloud | 0.2 units/bps |
| The backhaul cost paid by SBS edge cloud | 0.1 units/bps |

Deep neural network(DNN) is the important part of the deep Q-learning algorithm, and in the following we will provide the parameters for the composition of DNN. The DNN is composed of a input layer, two hidden layers, and a output layer. Meanwhile, the first hidden layer is set as 120 hidden neurons, and the second has 80. The training interval is set as 10, and training batch size is 128. In addition, the learning rate is 0.005, and memory size is 1024.

*7.2. Simulation Result*

The influence in simulation result by parameters will be investigated in the following experiment. Figures 9 and 10 presents the changing reward value and learning rate by our scheme with $UE = 20$. From Figure 9, we can obtain that with the iteration increases, the reward value increases gradually at the first 400 iteration, and then retain stable after convergence. As shown in Figure 10, the training loss decreases rapidly for the first 100 iterations and then remains unchanged. In conclusion, our proposed scheme is capable of achieving the near optimal solution relying on its exploration.

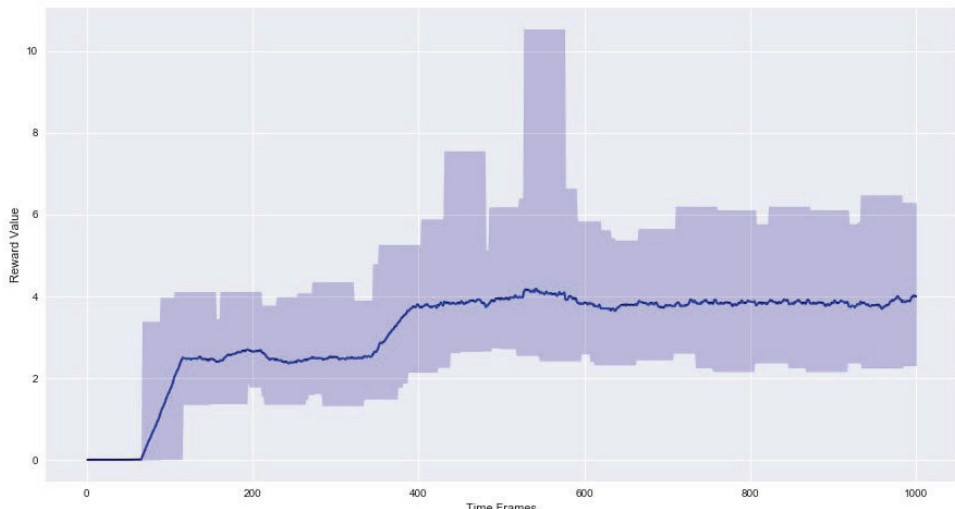

**Figure 9.** Reward value changes versus iterations.

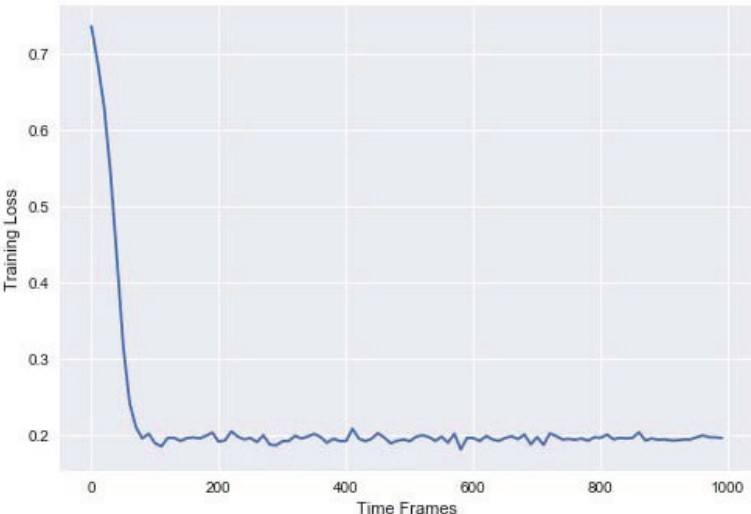

**Figure 10.** Learning rate changes versus iterations.

In this subsection, we have provided the impact on simulation result by three vital overhead parameters in DRL, i.e., learning rate, batch size, and training interval. Learning rate controls the learning process of the model. To be specific, a large learning rate may result in a rapid convergence, while a small learning rate is likely to bring over-fitting. Batch size affects the optimization degree and speed of the model. If the value of batch size is too small, the algorithm does not converge in epochs. Otherwise, it is easy to fall into local convergence. Similarly, training interval is also an important parameter in deep neural networks.

In Figure 11, the learning rate is set as 0.005, 0.01, 0.05, or 0.1. Specifically, when learning rate equals 0.01, the reward value required by DRL algorithm performs better than others during the whole iterations. Learning rate = 0.1 can not achieve an effect such as learning rate = 0.01, but has a better performance than other two cases. In addition, learning rate = 0.05 has the worst performance.

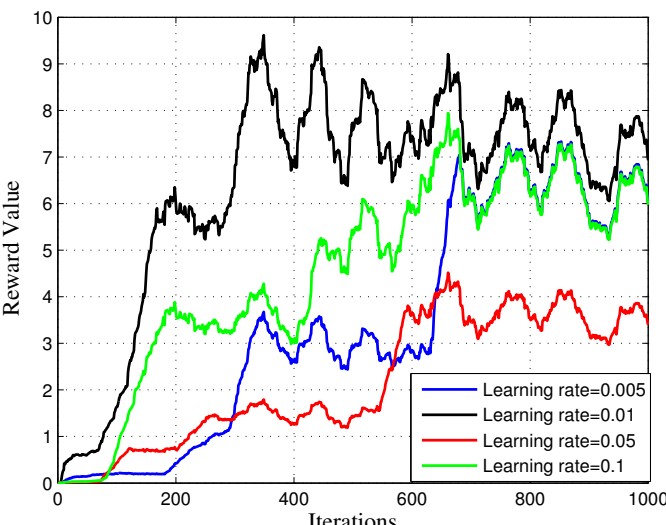

**Figure 11.** Reward value changes with different learning rate.

As shown in Figure 12, the batch rate is set as 32, 64, 128, or 256. Specifically, when batch size equals 32, the reward value required by DRL algorithm does not perform well in the at the first stage. However, it increases rapidly and achieves a best performance in the latter stage. *Batch size* = 64 has a slightly worse performance than the case of

*batch size* = 32. Additionally, *batch size* = 128 and *batch size* = 256 are not capable of performing well.

Figure 13 plots the reward value changes against iterations with different training intervals. To be specific, training interval is set as 5, 10, 15, or 20. Obviously, we can see that when training interval is 32, the reward value required by DRL algorithm performs well during the whole stage. When *training interval* = 5, 15, 20, the reward values do not have a well performance. Furthermore, *training interval* = 5 performs worst among these.

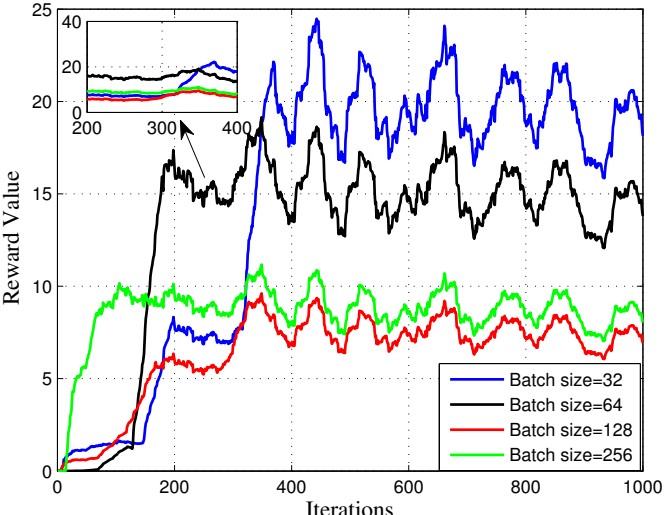

**Figure 12.** Reward value changes with different batch size.

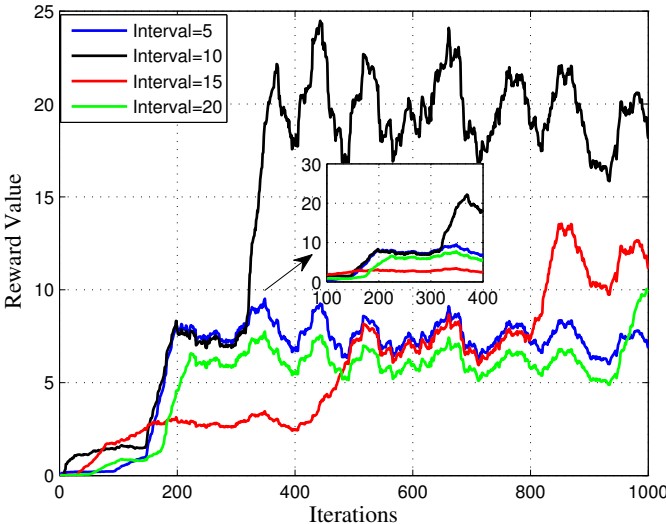

**Figure 13.** Reward value changes with different interval.

In order to verify the effectiveness of the calculation offloading strategy proposed in this paper, the genetic algorithm (GA), particle swarm algorithm (PSO), and bat algorithm (BA) are used for testing and comparison. The algorithms above are classical bionic optimization algorithms with fast convergence abilities. The genetic algorithm [26] is a method to search the optimal solution by simulating the natural evolution process. Particle swarm optimization [27] is a random search algorithm based on group cooperation, which is developed by simulating the foraging behavior of birds. The bat algorithm [28] is a method that simulates the movement process of individual bats and searches for prey to solve problems. The number of particles in each algorithm is set to 20, and the number of iterations is set to 50.

Figure 14 shows the change of UE randomly distributed in the network area. Specifically, the number of UAV equipment is set to [5, 10, 15, 20, 25, 30, 35, 40, 45, or 50] to test the influence of the number of equipment on the experimental results. Compared with the other three algorithms, DRL is able to obtain the best optimal value under different number of UEs. In contrast, the performance of the particle swarm optimization algorithm is slightly inferior to that of the bat algorithm. In addition, the genetic algorithm has the worst performance in all figures. Relatively speaking, more UE numbers require greater computational complexity.

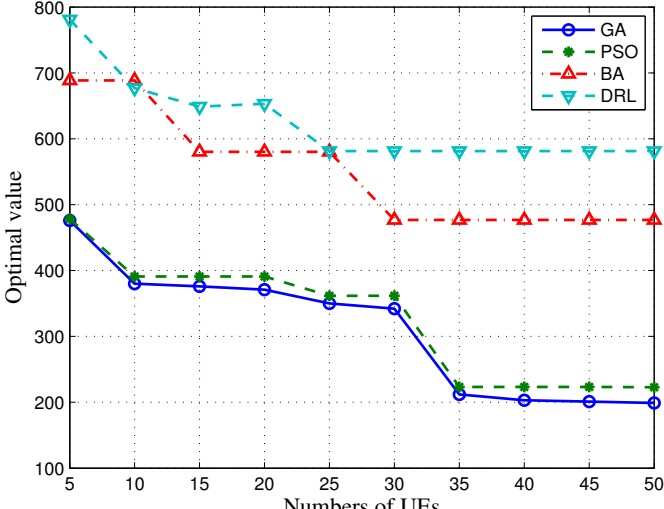

**Figure 14.** Comparison with other algorithms.

## 8. Conclusions

Aiming at the application of novel technologies in mobile edge computing, this manuscript proposed an intelligent computing offload and resource allocation edge computing technology. In the network architecture of the system environment, the proposed algorithm can adaptively provide effective edge computing strategies according to the communication resources, computing resources, and cache resources in edge computing. The simulation results show that the scheme can quickly converge to a satisfactory solution and verify the influence of different parameters of the scheme on the experimental results.

**Author Contributions:** Writing-review and editing, F.L.; Methodology, C.F.; Software, M.L.; Investigation, N.L. and T.S. All authors have read and agreed to the published version of the manuscript.

**Funding:** This research received no external funding.

**Conflicts of Interest:** The authors declare no conflict of interest.

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
