# Peer review of "Intelligent Computation Offloading Mechanism with Content Cache in Mobile Edge Computing"

_electronics, doi:10.3390/electronics12051254_

Round 1

Reviewer 1 Report

The connection between sections 5 and 6 is not clear. In essence they both refer to resource allocation. Why do the authors have different sections for seemingly similar algorithms, and what is the connection of one to another? Please add some more explanations in the manuscript, and consider rearranging the sections.

The whole evaluation section 7 seems very weak. The authors present some figures of certain metrics by varying some RL parameters. There is no insights on the performance of the allocation algorithms, and a comparison to other approaches.  

Line 430. The authors mention the parameters of a DNN. What is the purpose of this DNN? This hasn't been mentioned before in the manuscript.

English needs to be revised in the largest part of the manuscript.
Line 13. Describe/Mention these challenges.
Line 14. Describe/Mention these scenarios.
line 244, 252 revenue instead of avenue?

Reviewer 2 Report

In this paper, the authors explored an efficient computation incentive mechanism with the appropriate task offloading and resource allocation strategies. In this paper, they proposed an intelligent computation offloading mechanism with content cache in mobile edge computing. Experiment results demonstrate the effectiveness of the proposed scheme. However, I still have some concerns as follows.

1. The data structure and data model of the system should be clearly described.

2. What are the overheads (e.g., system setup, data storage, computing efficiency, etc.) of the proposed scheme?

3. The authors have claimed the offloading with the static analysis of edge computing. How to differentiate the authors’ work from these works which are already performed in distributed scenarios? More discussion should be added in the paper.

- Make smartphones last a day: Pre-processing based computer vision application offloading

-Smartphone-assisted energy efficient data communication for wearable devices.

4. There are more opportunities for conducting meaningful experiments to comprehensively evaluate the system performance and overheads.

5. More technical papers about edge computing should be investigated and analyzed. For example:

- SCoP: Smartphone energy saving by merging push services in Fog computing

- Smartphone-assisted smooth live video broadcast on wearable cameras

- An efficient learning-based approach to multi-objective route planning in a smart city

- Completion time and energy optimization in the UAV-enabled mobile-edge computing system

- Energy-efficient UAV-assisted mobile edge computing: Resource allocation and trajectory optimization

Round 2

Reviewer 1 Report

The authors did not address the comment about the performance of the allocation algorithms and the comparison of their proposal to other approaches.

The results section is very weak.

Reviewer 2 Report

All problems have been solved.

Round 3

Reviewer 1 Report

The authors addressed my comments and the manuscript can be accepted. Some references and short explanations of the algorithms in line 476 (GA, etc) should be provided.
